# Systematic Idea Refinement for Machine Learning Research Agents

**Zijun Liu**[1,2]**, Cheng Gao**[1]**, Hanxi Zhu**[1]
[1]Dept. of Comp. Sci. & Tech., [2]Institute for AI
Student ID: 2024310673, 2024210899, 2024310666

## Abstract

This project aims to enhance the machine learning (ML) research capabilities of large language model (LLM)-empowered agents beyond basic code generation. While recent advancements have demonstrated the use of single-agent systems for generating code across various ML tasks, these methods often focus solely on improving code validity. They lack the ability to explore diverse methodologies for a given problem, which limits their adaptability and performance. To address this gap, the proposed project develops a multi-agent framework that systematically refines research idea guidelines through automatic proposal, feedback integration, and inference-time scaling. By incorporating multi-level feedback from LLM judgments, code generation processes, and experimental results, this approach enables agents to explore a broader range of solution pathways, similar to human researchers. The framework is plug-and-play on code generation agents, and will be evaluated on *a total number of 75 Kaggle competitions*. The expected outcome is an improvement in the understanding and performance of machine learning research agents through a comprehensive exploration of methodological ideas.

## 1 Background

The rapid advancement of large language models (LLMs) has opened new avenues for automating machine learning (ML) research, and LLM-empowered agents are increasingly being applied to automate and augment research processes, ranging from generating code for general ML models [1, 2] to designing and executing data analysis pipelines [3]. By leveraging the generative and reasoning capabilities of LLMs, these agents are poised to transform the landscape of ML research, making it much more efficient and accessible on causal applications with less professional users.

However, despite the promise of LLM-empowered agents, their current applications remain limited in several key aspects. Most notably, existing approaches often focus on generating code snippets that solve specific tasks based on a single predetermined or self-determined methodology. For example, agents may be determined to solve a classification problem by generating code for a logistic regression model, without noticing it is not proper for some dense temporal data they are facing. This constraint stems from the fact that current LLM-based agents tend to reflect the initial idea or prompt that guides their code generation process. In practice, agents may consistently default to commonly used techniques, resulting high validity rates but rather low performance [4]. Techniques like tree searching [5] or iterative improvement [6] are employed to ensure the syntactic correctness of the generated code, but fall short when it comes to exploring a wider array of potential approaches for the underlying problem, which can severely limit the agents' performance, particularly where the optimal solution may not be apparent from the outset.

## 2 Task & Framework Formulation

**Machine Learning Research Problems** The core problem addressed in this project is to conduct ML research. Formally, let $\mathcal{D}_k = \{(x_i, y_i)\}_{i=1}^{N}$ denote the $k$-th benchmark of a given ML problem, where $x_i$ represents input features and $y_i$ represents the corresponding labels. The goal is to identify a model $f : \mathcal{X} \to \mathcal{Y}$ that maximizes a metric function $\mathcal{L}(f(x), y)$ over all benchmark $\mathcal{D}_k$. For a

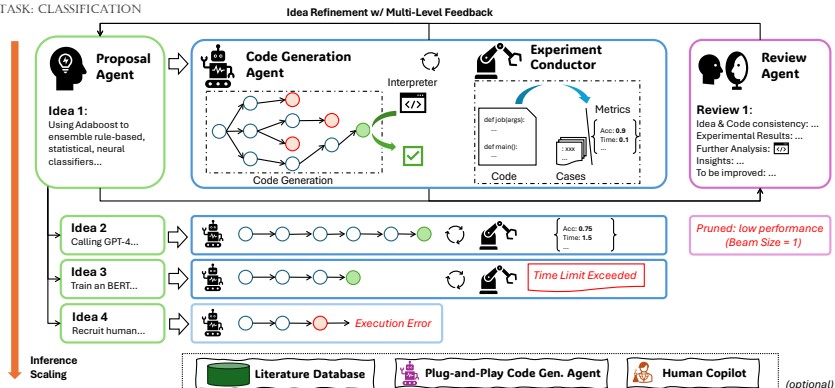

Figure 1: The overview of our proposed multi-agent idea refinement framework for ML researches.

given ML problem $P$, LLM-based agents are assigned to generate models $f \in \mathcal{F}$, representing a potential solution using diverse methodologies. It is worth noting that agents could use a training set for method development and validation, and could access the metric $\mathcal{L}$. Thus, multiple candidates $\mathcal{F}' \in \mathcal{F}$ could be generated after multi-sampling, and a best outcome could be selected $\tilde{f} \in \mathcal{F}'$.

**Single-Agent & Multi-Agent Systems** In a single-agent system, let $A_{\text{single}} : \Pi \rightarrow \mathcal{F}$ be an LLM-empowered agentic workflow that generates a model $f \in \mathcal{F}$ from coding based on an initial prompt $\pi \in \Pi$ that describes the task $P$. The generated model $f$ has the objective to maximize the metric function $\mathcal{L}$. A multi-agent system $\mathcal{M} : \Pi \rightarrow \mathcal{F}$ forms communications between a set of agents $\mathcal{A} = \{A_1, A_2, \ldots, A_m\}$, where each agent $A_i$ is specialized with its input prompt $\pi_i$, tools, and the workflow. The agents collectively propose and refine diverse methodologies for solving $P$ by exchanging messages between each other in a predefined order. In this project, we focus on designing plug-and-play methods for code generation agents, which means only one agent generate the final code and thus produce the ML model, and others mainly provide guidelines or feedback.

## 3 Related Work

**Machine Learning Research Agents** LLM-based agents have become integral to automating ML researches [7]. These agents leverage the generative capabilities of LLMs, using methods like multi-agent workflow [2], tree search [5], or iterative refinement [6] to improve code validity and correctness. These systems can produce syntactically correct and functionally valid solutions. However, recent works [4] have demonstrated that such systems can be effective in producing valid solutions across a range of problems while failing to yield higher performance. This underscores the need for frameworks that can propose and refine ideas to guide code generation for general ML tasks. Still, all systems above may face performance degradation when facing poorly defined tasks, e.g., without clear metrics, which we decide to leave for future works.

**Research Idea Generation** The concept of research idea generation extends beyond coding, aiming to replicate the behavior of human researchers. However, most existing approaches [1, 8] focus on refining an initial idea with inherent knowledge in LLMs or existing literature, rather than taking richer feedback from experimental results. In contrast, human researchers often select the most promising approach based on both theoretical and empirical results. Emulating this idea-driven exploratory process has the potential to enhance the effectiveness. The lesson of previous works [9] show that the integration with code generation agents might be the key challenge.

## 4 Proposed Method

To overcome above challenges, a multi-agent approach is proposed as a means to systematically explore and refine research ideas. As shown in Figure 1, the proposed framework integrates idea proposal, multi-level feedback, and inference-time scaling to enhance the diversity and quality of the solutions. The plug-and-play nature of the framework means it can be adapted to work with various code generation agents. The framework will be evaluated on MLE-Bench [4], which incorporates 75 Kaggle competitions. The proposed method could be applied on different code generation agents with slight effort, e.g., on tree searching or iterative improving agents. The baseline methods will be the current state-of-the-art idea generation agent systems [1, 8]. All methods will be plugged with a same code generation agent, and the performance will be compared on the same dataset. We wil also investigate in inference-time searching based on the metric $\mathcal{L}$ as clear rewards for different methods.

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
