# OpenReview forum: "【Proposal】Systematic Idea Refinement for Machine Learning Research Agents"
_tsinghua.edu.cn/THU/2024/Fall/AML — THU 2024 Fall AML Submission_

### Official Review · ~Iat_Long_Iong1 · 2024-11-07
**Good proposal**

**Rating:** 10
**Confidence:** 4

**Review:**

The proposed multi-agent framework for systematic idea refinement in machine learning research demonstrates significant potential to revolutionize the field. By moving beyond code generation validity and embracing diverse methodologies, this approach enables LLM-empowered agents to explore a broader range of solutions, similar to human researchers. The integration of multi-level feedback ensures a dynamic and iterative learning process, leading to more informed decision-making and potentially groundbreaking discoveries.

With its plug-and-play design and focus on real-world applicability, this framework holds the promise of transforming machine learning research, making it more efficient, accessible, and impactful for a wider audience.

---

### Official Review · ~Feihong_Zhang1 · 2024-11-11
**Systematic Idea Refinement for Machine Learning Research Agents**

**Rating:** 8
**Confidence:** 3

**Review:**

Review Comments
This paper proposes an innovative multi-agent framework aimed at enhancing the adaptability and diversity of large language model (LLM)-driven research agents for machine learning. By incorporating systematic research idea generation and a multi-level feedback mechanism, the framework provides guidance to code generation agents, enabling them to tackle a broader range of machine learning tasks. The work presents both theoretical novelty and practical applicability, with a clear structure and logical flow. Here are my detailed comments on the paper:

Strengths
Innovation: The authors introduce a multi-agent system to support research idea generation and a multi-level feedback mechanism, surpassing existing single-agent approaches. The framework’s feedback mechanism and methodological diversity promise improved adaptability when tackling complex tasks.

Technical Details and Feasibility: The paper provides a well-defined description of the framework’s components, including the Proposal Agent, Code Generation Agent, Review Agent, and Experiment Conductor, clearly outlining each module’s role. This clarity offers a strong foundation for practical implementation.

Evaluation Strategy: The authors plan to evaluate this framework across 75 Kaggle competition tasks and compare it with existing methods. Such empirical evaluation offers a robust means of validating the effectiveness and generalizability of the framework.

Suggestions for Improvement
Experimental Design Details: While the paper mentions that the framework will be tested across several Kaggle tasks, it lacks specific discussion on evaluation metrics such as accuracy or runtime. Including a more detailed performance comparison across different agent combinations would improve result credibility.

Potential for Further Expansion: Currently, the framework mainly targets code generation agents. The authors are encouraged to explore this framework’s applicability to other areas, such as data processing or feature engineering, to demonstrate its versatility within broader machine learning workflows.

Discussion on the Feedback Mechanism: Although the multi-level feedback mechanism is compelling, there is limited discussion on balancing feedback weights or dynamically adjusting feedback strategies. Further exploration of feedback mechanisms in future work could help optimize system performance.

Summary
Overall, this paper presents a highly innovative multi-agent framework that effectively enhances the adaptability and diversity of code generation agents through systematic idea generation and multi-level feedback. Although there is room for improvement in the experimental section, the framework design and implementation make a valuable contribution to the field. I recommend accepting this paper and look forward to seeing its broader applications and advancements in future research.

---

### Official Review · ~Mingdao_Liu1 · 2024-11-11
**Review for "Systematic Idea Refinement for Machine Learning Research Agents"**

**Rating:** 10
**Confidence:** 4

**Review:**

The proposal aims to enhance the machine learning research abilities of LLM-agents through y incorporating multi-level feedback from LLM judgments and exploring a broader range of solution pathways. The authors plan to evaluate the proposed framework on 75 Kaggle competitions.

The proposal is well-written and includes all required parts for a proposal. The research question highlights an important setting applicable to realistic macine-learning scenerios. The proposed research plan is concrete and detailed, including methods, baselines, evaluation metrics and expected results.

---

### Official Review · ~Gausse_Mael_DONGMO_KENFACK1 · 2024-11-11
**Clear Motivations**

**Rating:** 8
**Confidence:** 3

**Review:**

This paper presents a novel approach for improving LLM-empowered agents in machine learning research by enhancing their capacity to generate and refine diverse methodologies. The framework introduces a multi-agent system where agents collaborate to generate, evaluate, and refine research ideas through multi-level feedback, emulating human research processes more closely.

strengths : The authors identify the limitations of current LLM-based ML research agents; adressing theses limitations has the potential to enhance their generalization capabilities. The proposal of a multi-agent system with distinct roles could produce more diverse solutions.

weakness: Lack of details about the feedback mechanisms and the final agent generating the code.

---

### Official Review · ~Jiajun_Xu3 · 2024-11-11
**Great Proposal**

**Rating:** 10
**Confidence:** 4

**Review:**

The proposal presents an intriguing approach to enhancing the capabilities of large language model (LLM)-empowered agents in the domain of code generation. The authors have identified a significant gap in current LLM applications, which tend to focus on code validity rather than exploring diverse methodologies for solving problems. The proposed multi-agent framework aims to fill in the gap.

With solid foundations on the introduction to the problem and relevant researches, the proposal outlines a comprehensive framework that incorporates idea proposal, feedback integration, and scaling. And its plan to evaluate the framework on MLE-Bench, which includes 75 Kaggle competitions, provides a rigorous testbed for the proposed methods.

---

### Official Review · ~Jiaxiang_Liu7 · 2024-11-11

**Rating:** 8
**Confidence:** 4

**Review:**

This proposal presents a multi-agent framework aimed at enhancing the research idea generation and refinement capabilities of machine learning agents. The proposed system combines idea proposal, feedback integration, and inference-time scaling to tackle the limitations of single-agent models, particularly in exploring diverse methodologies. By using multi-level feedback from LLM-based judgments, code generation, and empirical results, the framework allows agents to refine solutions systematically, akin to human researchers. The methodology is well-delineated and shows promise for improving ML research automation, though further elaboration on handling computational efficiency and ensuring the quality of generated ideas would strengthen its practical feasibility. Overall, this proposal addresses a crucial aspect in advancing LLM-based agents for complex research tasks and holds significant potential for impactful contributions.

---

### Official Review · ~KAI_JUN_TEH1 · 2024-11-11
**Clear problem statement and technical approach.**

**Rating:** 10
**Confidence:** 5

**Review:**

This project aims to enhance the machine learning (ML) research capabilities of large language model (LLM)-empowered agents beyond basic code generation. The proposed multi-agent framework systematically refines research idea guidelines through automatic proposal, feedback integration, and inference-time scaling. Firstly, the text identifies the reasons for the agents' insufficient performance and suggests avenues for improvement. Then, it introduces the mechanisms of both single-agent and multi-agent systems. Finally, it presents some innovative and feasible research proposals. I wish you success!

---

### Official Review · ~Suraj_Joshi2 · 2024-11-12
**Review on Proposal For ML Research Agents**

**Rating:** 10
**Confidence:** 5

**Review:**

This project aims to expand the capabilities of LLM empowered agents beyond generating just a solution, that might not be optimal by using multiple agents. This project also aims to solve the problem of lack of innovation in output of LLM based agents that is heavily dependent on the initial prompt supplied. The "plug-and-play" nature of the code generation agent, designed to integrate easily with existing systems, is a practical feature that could facilitate broader adoption and adaptability.

Overall, this a very good proposal, everything is clearly articulated. and the only thing I doubt about the project is if there are multiple agents exploring various ideas, and as the LLMs are non-deterministic models, sometimes small change in output of one agent can lead to highly inconsistent outputs.

---

### Official Review · ~Wanlan_Ren1 · 2024-11-12
**Review for "Systematic Idea Refinement for Machine Learning Research Agents"**

**Rating:** 9
**Confidence:** 4

**Review:**

This proposal presents a promising approach to enhance machine learning research agents by developing a multi-agent framework that enables systematic idea refinement. The project’s emphasis on expanding LLM-driven agents beyond simple code generation is valuable, as it addresses current limitations in adaptability and performance. Strengths include the innovative use of multi-level feedback to diversify solution pathways and the plug-and-play compatibility with various code generation agents. However, further clarity on the metrics for evaluating the agents’ performance would strengthen the proposal. Overall, this work has the potential to significantly improve LLM-driven ML research and provides a solid foundation for advancing automated research processes.

---

### Official Review · ~Kittaphot_Saengprachathanarak1 · 2024-11-12
**Review of "Systematic Idea Refinement for Machine Learning Research Agents"**

**Rating:** 9
**Confidence:** 4

**Review:**

The paper presents a well-structured multi-agent framework for ML research agents, integrating multi-level feedback from LLM judgments, code generation, and experiments to enhance solution diversity beyond single-agent systems. It stands out for its originality, emulating human-like research exploration to address the limitations of prior code-focused methods. The plug-and-play design adds versatility, allowing integration with different code generation agents across various ML tasks. While the paper’s clarity and organization are strong, further empirical validation and detail on agent communication protocols would strengthen its contributions. Therefore, this innovative framework offers promising advancements for adaptable, autonomous ML research tools.

---

### Decision · Program_Chairs · 2024-11-18

**Decision:**

Strong Accept (Long Presentation)

**Comment:**

**Systematic Improvements in Machine Learning Research Agents**

**2.3.1 Key Innovations**
1. Introducing multi-agent techniques for automated machine learning
2. Developing workflows inspired by human researchers

**2.3.2 Additional Key Information**
None

**2.3.3 Advantages**
1. Creative and innovative topic selection

**2.3.4 Areas for Improvement**
1. Explore the current advancements of multi-agent systems in automated machine learning tasks
2. Clarify how performance is evaluated, as executable feasibility and efficient execution may require distinct evaluation metrics

**2.3.5 Recommendations**
1. Define the task boundaries clearly, breaking them down into more specific and actionable components to facilitate focused research in a shorter time frame.